# Advanced eNose-Driven Pedestrian Tracking Pipeline for Intelligent Car Driver Assisting System: Preliminary Results

**DOI:** 10.3390/s22020674

**Published:** 2022-01-16

**Authors:** Francesco Rundo, Ilaria Anfuso, Maria Grazia Amore, Alessandro Ortis, Angelo Messina, Sabrina Conoci, Sebastiano Battiato

**Affiliations:** 1STMicroelectronics, ADG Central R&D, 95121 Catania, Italy; ilaria.anfuso@st.com (I.A.); mariagrazia.amore@st.com (M.G.A.); angelo.messina@st.com (A.M.); sabrina.conoci@unime.it (S.C.); 2Department of Mathematics and Computer Science, University of Catania, 95125 Catania, Italy; ortis@dmi.unict.it (A.O.); battiato@dmi.unict.it (S.B.); 3Institute for Microelectronics and Microsystems (IMM), National Research Council, 95121 Catania, Italy; 4Department of Chemical, Biological, Pharmaceutical and Environmental Sciences, University of Messina, 98166 Messina, Italy

**Keywords:** driver safety, alcohol detection, artificial neural networks

## Abstract

From a biological point of view, alcohol human attentional impairment occurs before reaching a Blood Alcohol Content (BAC index) of 0.08% (0.05% under the Italian legislation), thus generating a significant impact on driving safety if the drinker subject is driving a car. Car drivers must keep a safe driving dynamic, having an unaltered physiological status while processing the surrounding information coming from the driving scenario (e.g., traffic signs, other vehicles and pedestrians). Specifically, the identification and tracking of pedestrians in the driving scene is a widely investigated problem in the scientific community. The authors propose a full, deep pipeline for the identification, monitoring and tracking of the salient pedestrians, combined with an intelligent electronic alcohol sensing system to properly assess the physiological status of the driver. More in detail, the authors propose an intelligent sensing system that makes a common air quality sensor selective to alcohol. A downstream Deep 1D Temporal Residual Convolutional Neural Network architecture will be able to learn specific embedded alcohol-dynamic features in the collected sensing data coming from the GHT25S air-quality sensor of STMicroelectronics. A parallel deep attention-augmented architecture identifies and tracks the salient pedestrians in the driving scenario. A risk assessment system evaluates the sobriety of the driver in case of the presence of salient pedestrians in the driving scene. The collected preliminary results confirmed the effectiveness of the proposed approach.

## 1. Introduction

In automotive applications, there are significant interests in the development of innovative technologies to increase the level of safety. A wide variety of sensing devices that show a robust ability in monitoring the driver’s attentional status are being implemented in cars [1,2]. The issue of driving safety has been significantly explored by researchers in the scientific field. Automotive statistics clearly highlighted a significant number of road accidents involving pedestrians [1,2,3,4,5]. Among the aspects that most negatively affect driving safety, there is certainly a poor level of driver attention, both physiological and correlated to an altered state of alertness, often linked to alcohol abuse. For this reason, scientific research has contributed to the development of several innovative solutions that allow intelligent monitoring of the driver’s attention level associated with a correlated driving assistance (so-called Advanced Driver Association Systems (ADAS) solutions) [6,7,8,9,10,11]. More in detail, scientific research has largely investigated primarily the development of the hardware platform (sensors, microcontrollers, interconnection systems, etc.) capable of hosting the driver monitoring and driving assistance algorithms as well as supporting the complex computational processing of data (visual and numerical) deriving from the characterization of the automotive environment. Subsequently, the researchers focused their investigations on the development of efficient algorithms mostly based on artificial intelligence, which would allow for robust data processing with a consequent definition of an accurate near-real-time response [1,2,3,4,5,6,7,8,9,10,11,12]. Here are some details of the main development trends in the automotive field.

Such interesting solutions regarding the intelligent detection and analysis of the car driver heart rate dynamic through the PhotoPlethysmoGraphy (PPG) were investigated by the authors of the work herein exploited [3,4,5,6]. Further, Computer Vision-based solutions have been implemented by the authors and applied successfully in different automotive scenarios, specifically in the field of Advanced Driver Assisted Systems (ADAS) [7,8,9,10,11]. Anyway, the characterization of the driver’s level of attention can also be obtained by analyzing the breath of the subject driving [12]. As introduced, the car driver’s attentional status can be retrieved by an ad-hoc sober level assessment. As highlighted below, the subject’s drowsiness and sobriety level are closely correlated, as a high concentration of alcohol induces a proportional state of drowsiness [8,9,10,11,12,13].

To track the car driver’s sobriety and correlated drowsiness, an ad-hoc sensing framework has to be designed and implemented. Alcoholic levels above standard concentration limits (e.g., 0.08% g/mL in the USA or 0.05% in Italy’s legislation) can induce a state of dangerous drowsiness, affecting correlated body movements and reaction capabilities [8,9,10,11,12,13]. In this context, a strong boost to the scientific research development was given from the advent of strategic alliances made by car makers, industries, institutions and research bodies, with the common goal of developing robust and efficient solutions to reduce accidents due to alcohol abuse by those who start driving. In the context, it is worth mentioning the Driven Alcohol Detection System for Safety Alliance (DADSS) [13], which, among other research, has developed technical guidelines for the intelligent monitoring of the driver’s alcohol level/sobriety [13]. Clearly, a crucial role in this context is therefore played by the alcohol sensor, which must provide an accurate assessment of the driver-subject’s alcohol level in order to determine the subsequent actions to protect driving safety. For this reason, the authors introduce a brief description of the main sensing systems that can be used in these automotive application scenarios.

The first type of sensing device is the non-selective and general-purpose air-quality sensor, the so-called VOC (Volatile Organic Compounds) sensing devices. These inexpensive devices are activated in the presence of any volatile compound in the passenger compartment of the car, effectively producing a simple assessment of the air quality and certainly not of the driver’s alcohol level [13].

On the other hand, selective alcohol or ethanol sensors, although much more precise, are more expensive and therefore often incompatible with the development criteria in the automotive field [13]. To address the aforementioned problem, the authors have developed an algorithm based on Artificial Intelligence (AI) that is able to analyze the data sampled by a classic VOC sensor, identifying the specific features of the alcohol analyte from the embedded features to the acquired time series. In this way, by means of an AI-augmented low-cost sensing combination system, we are able to make a classic VOC sensor selective. We have tested our device for alcohol, but theoretically, it can be extended to any analyte.

As introduced, driving safety and road accidents involving pedestrians represent an important problem to address in the automotive field. For this reason, in addition to proposing an efficient and innovative solution for monitoring the sobriety (and related drowsiness) of the driver, the authors propose a contextual system of identification and tracking of the salient pedestrians in the driving scene. In this way, the full pipeline will be able to evaluate an overall level of risk correlated both to the state of attention and sobriety of the driver and to the level of riskiness of the driving scene, which may or may not include pedestrians and/or salient pedestrians. The concept of a “salient” pedestrian, which has a role in determining the level of risk (different from the concept of a pedestrian simply present in the driving scene but not directly involved in the risk assessment), will confirm to be an innovative imprinting for the proposed pipeline.

Specifically, we have correlated this intelligent sensing system to the issue of robust pedestrian tracking in the automotive field. As highlighted, driving scenarios that include pedestrians are particularly complex [11]. Several researchers are investigating the design of intelligent algorithms that explore the relationship between the driver’s attention level or sobriety to the presence of any pedestrians in the driving scene [10,14,15]. Pedestrian detection is a crucial task in the smart driving field. The main solutions are based on the analysis of visual information by exploiting complex deep neural networks. However, although image-based detection technology has made great progress in the last few years, some works suggest that multisensor fusion technology may improve the effect of pedestrian detection technology in practical applications [16]. In this context, the authors proposed a self-attention deep enhanced Mask RCNN (Region-based Convolutional Neural Network) network for the identification and tracking of the salient pedestrian in the driving scene guided by a sobriety monitoring system.

The use of a deep learning-based approach that monitors only the salient pedestrians (and not simply all the pedestrians embedded in the driving scene) associated with an intelligent system that selectively retrieves the level of sobriety (and therefore the correlated drowsiness) will allow a continuous monitoring of the driving risk level which, unlike the pipelines produced in the literature, will be both sustainable from a hardware point of view and efficient and accurate as confirmed by the performances reported in the “Experimental Results” section. For both of the introduced sub-systems, we briefly introduce the state of the art, and then we proceed reporting the implementation details of the proposed pipeline.

## 2. Related Works

Several researchers have investigated the development of such AI-based solutions for obtaining selective classical VOC sensors as well as investigated innovative materials to be used as a “sensing filter” for specific analytes, such as alcohol [17,18,19].

In [20], a novel technique using response characteristic curves was proposed. The data obtained from rise-time, peak-time and recovery-time were used as representatives for the characteristics of the response curves. The proposed system consisted of three semiconductor gas sensors. Even though the sensor was able to detect alcohol, it was not selective and also provided an output for other organic compounds, such as acetone, ammonia, etc. In [21], an optical gas sensor that requires low electrical power was designed. The proposed sensor was a Magnesium-tetraphenylporphyrin thin film device. The reported results are very promising. In [22], the authors implemented a surface-modified TiO_2_ nanoflower hybrid sensing device, incorporating Pd and rGO as a secondary material for surface enhancement. They tested the designed device for monitoring ethanol and methanol analytes, collecting very interesting performance results in terms of response magnitude towards methanol for the surface-modified nanostructures compared to its pristine counterpart. The synergistic effects of the noble metal catalyst Pd and 2D material rGO with pristine TiO_2_ nanoflower structure make these types of surface-modified binary composites as a potential alcohol sensor device. In [23], a sensor device embedding a low-cost light emitting diode (LEDs) array and a CMOS (Complementary metal–oxide–semiconductor) photodetector was proposed to analyze the color change in the sensing material. The sensing materials were tested with various common VOCs, such as alcohols, acetone, ammonia and water. Pattern recognition through the classical principal component analysis (PCA) was applied. The benchmark results showed promising performance but low ability to be selective to a single analyte. In [24], deep learning was used to perform a soft-sensing device for tracking alcohol analytes. Specifically, the authors proposed an artificial neural network single-layer perceptron (ANN-SLP) to process specific VOC datasets in order to determine multiple classifications of alcohol types. They obtained that the highest performers were the QCM3 sensor and QCM6 (QCM: quartz crystal microbalance—near 100% accuracy) in the sensing of different types of alcohol.

With the recent development of deep learning techniques, object detection has made great progress. In the context of intelligent driving, pedestrian detection has a crucial role, as it significantly affects drivers’ and pedestrians’ safety. Although a pedestrian detection task could be addressed as a general object detection problem, the specific task presents additional issues.

In general, there are two main approaches in deep learning-based detection: one-stage or two-stage detection. In the two-stage detection, first a number of region suggestion boxes are detected. Then, a predictor is applied on such regions. The one-stage detection aims to directly predict the detected object area, providing the final prediction result. Although the two-stage approach is more complex than the one-stage approach, it has better robustness and accuracy overall. Moreover, researchers can focus on the improvement of detection or prediction, providing benefits to the whole framework.

A significant step toward a real-time two-stage detector is represented by Faster RCNN [25].

The work in [26] proposed some improvements based on Faster RCNN and proposed a method named region-based fully convolutional networks (RFCN) [26], which improved the processing results of pedestrian detection in scenarios that are specifically sensitive to location information. Compared with Faster RCNN, the approach in [26] augments feature sharing, reducing redundancy in the architecture and improving its running speed. The RFCN [26] algorithm is mainly proposed for general object detection, and it can also achieve good results in specific pedestrian detection areas. In [27], the authors proposed a detector named Feature Pyramid Network (FPN) based on Faster RCNN. General purpose approaches place the detector after the extraction of a category-aware feature. Such an approach works well for general object detection. The work in [27] improved the pedestrian detection, proposing a top-down prediction structure based on high semantic information built on the whole convolution pipeline.

The Complexity Aware Cascade Training (CompACT) algorithm proposed in [28] optimizes classification and better combines feature extraction and a classifier function, which plays an important role in promoting pedestrian classification at different scales. The authors proposed the CompACT boosting algorithm for learning complexity-aware detector cascades, which are able to integrate multiple feature families, generalizing the two-stage detection approach. The CompACT algorithm shows high performances in the field of pedestrian detection, although it can be extended to other object detection tasks.

The Mask RCNN model proposed in [29] represents an improvement of Faster RCNN, which adds the extraction of a semantic segmentation mask. This improvement in the task of object detection caused a boosting effect also in the field of pedestrian detection. Indeed, this architecture allows the pedestrian segmentation and background separation, in addition to the basic detection of pedestrians [30].

The first one-stage deep-based detector named You Only Look Once (YOLO) has been presented in [31]. A single neural network is applied to the whole input, which is divided into regions. As a result, the detection speed is improved, and the region proposal is predicted jointly to detect probability. Most of the one-stage approaches are based on the YOLO detector and its improvements, thanks to their high computation speed. However, pedestrian detection methods mostly focus on the two-stage approach, given its higher accuracy rate in general.

Indeed, pedestrian detection adds specific requirements on accuracy and time performances, which is of high significance in the application of this specific task. In general, one-stage detection approaches represent a good choice if the main requirement is the computing time, while two-stage approaches show better prediction performances. Although some works suggest that one-stage detectors can improve their accuracy while keeping high speed [32], improving the detection rate and simultaneously maintaining the detection speed is currently a challenge.

Occlusion is a typical issue in pedestrian detection because people often move in crowds. This results into a significant limit of the application of current technology in smart driving.

The authors of [33] proposed a repulsive loss function named RepLoss to reduce the mutual influence between detected objects, with the aim to reduce the effect of occlusion in pedestrian detection. Although there have been attempts to improve the effect of pedestrian detection under occlusion [33,34], they involve an increase of computational costs and subsequent reduction of detection speed.

Another important issue related to the task of pedestrian detection is the presence of high-scale variability of the different pedestrians depicted in the scene. An example of attempt to address the multiscale problem in pedestrian detection is presented in [35], in which the authors proposed a method named Topology Localization and temporal feature aggregation (TLL), which integrates multiscale human body model information in the model. The use of deep models often involves high computational and storage requirements, which further limits the application on real-world scenarios. In addition, special real-world scenarios introduce additional problems (e.g., fog, rain, night, snow, etc.) that need to be addressed.

The detection and subsequent monitoring of pedestrians in the driving scene can help an automatic driver assistance system to validate instant by instant if the driving dynamics and the level of attention are compatible with the presence of pedestrians in the scene by combining the two sources of information. Many authors have investigated this relevant issue by analyzing the advantages inherent in the use of deep learning architectures [14,15]. In [14], the authors investigated such deep architectures to monitor and track the pedestrians. The performance was very promising. In [15], the authors proposed a solution named DeepParts, which can be trained on weakly labeled data, i.e., only pedestrian bounding-boxes without part annotations. DeepParts was confirmed as a good detector that can detect a pedestrian; extensive experiments confirmed that this approach outperformed the previous best method by 10%. In [36], the authors have used the YOLO framework for fast object detection combined with a MobileNet architecture for feature extraction and a set of algorithms to generate associations between frames, obtaining a performance of 93.2% in accuracy. In [37], the authors investigated the issue of vulnerable pedestrian detection, showing very interesting results applying such deep learning-based solutions. Anyway, most of the proposed solutions showed the issue to require more complex dynamics and data to be processed.

## 3. Methods and Materials

As introduced, the authors propose an advanced system that combines an alcohol selective soft-intelligent VOC sensor with a self-attention deep network for pedestrian tracking in a driving scene. An intelligent control panel provides a risk assessment for the pedestrian according to the detected sobriety related to the characterization of the tracked salient pedestrians. The following Figure 1 shows an overall scheme of the proposed pipeline:

As shown in Figure 1, the proposed pipeline is composed by two sub-systems: the Intelligent Soft-Sensing System and the Intelligent Pedestrian Tracking System. The output of both systems will be processed by an ad-hoc designed Intelligent Control Panel in order to retrieve a robust risk-assessment alert system.

### 3.1. The Intelligent Soft-Sensing System

The target of this system is to provide a robust assessment of the car driver’s alcohol-sobriety by using a classical VOC sensor with a downstream deep classifier. Specifically, we have designed a deep architecture that learns the embedded deep features of the car driver’s breath sampled through a prototype VOC sensor GHT25S developed by STMicroelectronics. As schematized in Figure 1, the GHT25S sensor will be hosted on the steering wheel base of the car at about 1 m from the driver subject. The collected car-driver breath data will be digitalized (ADC at 12-bit) and pre-processed by an automotive-grade microcontroller device (MCU) SPC58X Chorus, provided by STMicroelectronics [38]. The normalized data will be fed as an input to the deep architecture, which is ongoing to be ported to another higher performer and accelerated MCU, which is the STA1295 Accordo5, provided by STMicroelectronics [39]. The following Figure 2 reports a schematic of the so-designed sensing system.

More detail about the used sensing device are as follows. As introduced, gas sensors, such as a human sense organ like the nose, can capture atmospheric composition data, recognizing the presence of various types of gases and send this information in the form of an electrical pulse to the components that, like the human brain, can process them and return an output as required. In the scientific literature, one of the most studied gas sensors due to low cost and high sensitivity is the metaloxide resistive sensor (MOX). The used sensing devise GHT25S is a MOX sensor. In Figure 3, we report the overall internal schematic of the sensing device.

As reported in Figure 3, the GHT25S is a MOX sensor. On the base of the device, there is the substrate on which a specific integrated, dedicated MCU (ASIC) is hosted. The sensor is protected by a special metal cap provided to small holes to allow the continuous change of air to be analyzed in the internal small metal chamber. The operating temperature ranges from −40 °C to 85 °C (with +/− 0.6 °C). The technology of the embedded ASIC is HCMOS9A, 0.13 µm. The GHT25S sensor is a combination device, able to also detect humidity and temperature of the closed sensing scenario. The relative humidity (RH)ranges from 0 to 100% RH (+/− 3% RH). The sensing device shows a response time of <5 s, which is very close to automotive near-real-time requirements. Further details include: an operating heater power of 20 mW, Internal System resistance (Rs) from 1 kHom to 3 MHom and target sensing range from 0.5 to 100 ppm of TVOC equivalents. Finally, the GHT25S can be driven through an SPI (Serial Peripheral Interface) or I^2^C interface, and it needs a supply voltage of 1.7 to 3.6 V. Depending on the nature of the analyte (embedded in the driver breath) coming into contact with the GHT25S sensing layer, given the characteristics such as mass, temperature, humidity and electrical resistance of its particles, contact or absorption to the sensing layer produces a chemical reaction that generates a certain electrical impulse with the specific characteristics of frequency, current, voltage or impedance/conductance, enabling a specific biological signature of that analyte. The downstream post-processing system composed by the deep architecture will further process that car driver breath signature, trying to associate the extracted deep features with the type of the source analyte. The following Figure 4 shows the used GHT25S sensing device.

The proposed GHT25S sensor requires a minimal calibration as for all similar sensing devices [40,41,42]. In the following figures, we reported some instances of the sampled car driver breath data with the sensing device as per Figure 2, both in case of a sober subject and in case of a driver who drank. The data diagrams reported in Figure 5 show a time snapshot of the correlated sensing items, such as the ppm measurements, temperature (T), humidity (RH) and internal resistances (Rsense, Rair, RSC).

The above sensing data will be properly normalized and fed as an input of the deep residual network designed to extract embedded features to be associated with the type of the driver: Sober or Not Sober. More details about the implemented deep backbone are as follows.

We designed a Deep 1D Temporal Dilated Convolutional Neural Network (1D-CNN) suitable to classify the GHT25S normalized sensing patterns [31]. A temporal convolutional residual network that embeds a dilated causal convolution layer capable of acting on the temporal stages of each set of sensing data sequence [43,44,45] was implemented. The proposed 1D-CNN is composed of 24 residual blocks with a dilated convolution (3 × 3 kernel filters), followed by normalization, ReLU activation blocks and spatial dropout. The deep backbone includes a final softmax stage for data classification. For each of the blocks, there is a progressive increase in the dilation starting from 2 and increasing with a power of 2 until to 32. The output of the 1D-CNN is a binary assessment (0–0.5: Sober Driver; 0.51–1: Not Sober Driver) of the driver’s attentional level associated to the sampled breath sensing data. The described Deep Learning framework proved to be effective in assessing the driver’s level of sobriety with high precision and timing performance as shown by the results reported in the related section.

### 3.2. The Intelligent Pedestrian Tracking System

The authors investigated several interesting object detection and tracking architecture backbones to adapt to pedestrian tracking. However, we found it useful to implement an innovative network that included the recent self-attention context using Criss-Cross layers [45]. As schematized in Figure 1, an enhanced Mask-R-CNN architecture [29] embedding Self Attention is proposed. Mask-R-CNN is widely used in automotive applications [29]. The target of the implemented enhanced Mask-R-CNN is that it allows for the performance of a pixel-based segmentation of the input image representing the driving scene frame. Moreover, with this solution, we can generate the corresponding bounding-box that characterizes the Region of Interest (ROI) on which to perform post-processing.

As a feature generator backbone, we embedded a ResNet-101 [46]. The so-segmented road (ROI) will be fed as an input of the enhanced downstream classifier based on a further ResNet-101 backbone. This deep classifier embeds a Recurrent Criss-Cross Attention (RCCA) layer [46]. The attention mechanism based on the Criss-Cross algorithm was first proposed in [46], showing a very promising performance in computer vision tasks. Specifically, the proposed Criss-Cross attention module performs an innovative pixel-based contextual processing of the input image frame. More in detail, this algorithm leverages full frames embedding dependencies during the learning session of the deep network. Let us formalize the Criss-Cross algorithm. Given a local feature map **H**
**∈**
**R^C × W × H^**, where **C** is the original number of channels while **W × H** represents the spatial dimension of the generated feature map. The Criss-Cross layer applies two preliminary 1 × 1 convolutional processing to generate two feature maps **F_1_** and **F_2_,** which belong to **R^C^′^ × W × H^** and in which **C^′^** represents the reduced number of channels with respect to original **C**. The Affinity function is suitable to generate the Attention-Map **A_M_**
**∈**
**R^(H +W − 1) × (W × H)^**. The Affinity operation can be defined as follows. For each position **u** in the spatial dimension of **F_1_**, a vector **F_1,u_**
**∈**
**R^C^′^^** can be retrieved. Similarly, we define the set **Ω****u**
**∈**
**R^(H + W − 1) × C^′^^** by extracting feature vectors from **F_2_** at the same position **u**, so that **Ω****_i,u_**
**∈**
**R^C^′^^** is the i-th element of **Ω****u**. After these preliminary operations, we can define the introduced ***Affinity*** operation as follows:
(1)$$\delta_{i,u}^{A}=F_{1,u}\Omega_{i,u}^{T}$$
where $\delta_{i,u}^{A}\in D$ is the so-called ***Affinity function***, i.e., the degree of mathematical relation between features **F_1,u_** and **Ω_i,u_**, for each **i** in the range **[1, H + W − 1]**, and **D ∈ R^(H + W − 1) × (W × H)^**. Finally, we apply a softmax layer on the space **D** to determine the attention map **A_M_**. Finally, another convolutional layer with a 1 × 1 kernel will be applied on the feature map **H** to generate the re-mapped feature ***θ* ∈ R^C × W × H^** to be used for spatial adaptation. At each position **u** in the spatial dimension of ***θ***, we can define a vector ***θ*_u_ ∈ R^C^** and a set **Φ_u_ ∈ R^(H + W − 1) × C^**. The set **Φ_u_** is a collection of feature vectors in ***θ*** having the same row or column with position **u**.

At the end, the desired pixel-based contextual information is retrieved through the ***Aggregation*** functional re-mapping, defined as follows:
(2)$$H_{u}^{\prime }=\overset{H+W-1}{\underset{i=0}{\sum }}A_{M}^{i,u}\Phi_{i,u}+H_{u}$$
where $H_{u}^{\prime }$ is a feature vector in **H^′^ ∈ R^C × W × H^** at position **u** while $A_{M}^{i,u}$ is a scalar value at channel ***I*** and position **u** in the field **A_M_**. The so-defined contextual information $H_{u}^{\prime }$ is then added to the given local feature **H** to augment the pixel-wise representation and aggregating context information according to the spatial attention map **A_M_**. These feature representations achieve mutual gains and are more robust for semantic segmentation. Anyway, the Criss-Cross attention module is able to capture contextual information in horizontal and vertical directions, but the connections between the pixel’s neighborhood is not covered. To overcome this issue, the authors introduced a Recurrent Criss-Cross processing [33]. In the Recurrent Criss-Cross algorithm, each contextual operation can be unrolled into *R* loops. We defined *R* = 2 for our purpose as suggested by the authors [45]. We have embedded the so-described Criss-Cross layer (RCCA, i.e., Recurrent Criss-Cross Algorithm) to the latest residual block of the ResNet-101 backbone as shown in Figure 1.

More details on the reasons for which the authors preferred an architecture with Criss-Cross attention layers are as follows (thus confirmed by the scientific evidence obtained from the performed experimental tests):The Mask-R-CNN embedding Criss-Cross path modules produce more discriminative visual features;The so-designed enhanced attention-based network is a better performer with respect to the similar state-of-the-art solutions [45];It is GPU memory-friendly (significantly reduces Floating point Operations per Second (FLOPs) by about 85%) compared with the other attenion modules, such as the non-local block (it requires on average 11 × less GPU memory usage) [29,45];High computational efficiency.

Moreover, the main advantage of the proposed attention module is embedding in the usage of the contextual information. It is a common practice to aggregate contextual information to augment the feature representation in semantic segmentation or object detection deep architectures [45]. By means of the defined “Affinity” and “Aggregation” operators, we are able to collect contextual information in horizontal and vertical directions to enhance the pixel-wise representative capability and then the visual feature maps. In this way, we are able to perform a robust semantic segmentation of the detected and tracked objects (pedestrian in our case), as the pixel-based information is more accurate and discriminated with respect to the same and related to other pixel classes represented in the visual frames and features.

Through the contextual pixel-based information, we are able to train the deep architecture to efficiently detect domain-adapted objects, specifically, in our case, pedestrians in different scenarios while walking, by bike, on motorbike, etc. Moreover, ad-hoc training sessions that force segmentation of the only salient pedestrian subjects (i.e., the most important pedestrian from a driver visual point-of-view [8,9,10]), complete the innovative performance of the proposed pipeline, which will, therefore, be able to identify and segment salient pedestrians in different domain-adapted scenarios. The design of ad-hoc Mask-R-CNN architecture allowed us to obtain, in addition to the semantic segmentation, the bounding-box part of the segmented pedestrian also, which therefore also gives further spatial information related to the driving scene.

The following Figure 6 shows some instances of the tracked and segmented salient pedestrian. As introduced, the so-designed deep system has been trained to track only the salient pedestrians, leaving out those outside the salience scene, thus reducing the overall computational load of the pipeline. Furthermore, the so-enhanced Mask-R-CNN allows us to obtain the bounding-box of the pedestrian, which we will need to determine the distance from the driver’s car. Quite simply, the height and width of the segmentation bounding-box of each segmented pedestrian will be determined. Only bounding boxes that have at least one of the two dimensions greater than two heuristically fixed thresholds (L_1_ and L_2_, respectively, for length and width) will be considered relevant salient pedestrians, i.e., pedestrians that must be considered by the driver when choosing the driving dynamics. The other pedestrians will be considered non salient and, therefore, are not involved in the safety level assessment. This so-computed distance assessment will be used in the next block of the proposed pipeline.

As reported in Figure 6, the detected relevant pedestrian is thus classified since the bounding-box generated by the Mask-R-CNN network exceeds, in spatial-dimension, at least one of the L_1_ or L_2_ thresholds set for this pipeline. It is clear that this is a very close (spatially) pedestrian to the vehicle from which the visual perspective of the scene is supposed to be obtained. On the other hand, in Figure 6, a “not relevant” salient pedestrian is also identified since none of the dimensions of the bounding-box exceed the predetermined thresholds. In fact, this is a reasonably distant pedestrian and therefore not mainly involved in the risk assessment. Obviously, if the bounding-box dimension changes in subsequent time evolution (due to, for example, a vehicle or a pedestrian approaching), this pedestrian could be classified as relevant and therefore fall within the safety assessment described in the next section.

This architecture performed very well as we reached a test-set performance mIoU of 0.695 over a CamVid dataset, which is in line with the performance of other more complex architectures [29,30,31,32,33,34,35,36,37,38,39,40,41,42,43,44,45,46,47].

### 3.3. The Intelligent Control Panel

In the scheme reported in Figure 1, a block called Intelligent Control Panel (ICP) is highlighted, which will analyze the outputs produced by each of the previously described pipelines, specifically, the assessment of the driver’s sobriety according to the “relevant salient” assessment of the detected and segmented pedestrians. In detail, the ICP will trigger an acoustic alert signal with different intensities according to the risk level if one of the following setups becomes true:High-Risk Level (Alert-Signal with High intensity)

Detection of “Sober Driver” ***AND*** the Mask-R-CNN identifies such relevant salient pedestrians. Specifically, the high-risk condition is determined by the detection of a subject-driver who is not sober (has a blood alcohol level higher than the allowed threshold on which the sensor has been calibrated), associated with a driving scenario in which there are relevant pedestrians and, therefore, spatially in the salient risk area. Consequently, sound alarm alerts must be emitted by the proposed MCU systems described above in order to attract the driver’s attention. In the subsequent automation levels, in addition to the audible warning signal, the vibration of the car steering is also provided, and in the more advanced phases (not the subject of this work but under investigation of the authors), the progressive control of driving with autonomous actions taken by the embedding MCU’s algorithm, such as securing the vehicle and pedestrians in the driving scene.

Medium-Low Risk Level (No Alert)

Detection of “Not Sober Driver” ***AND/OR*** the Mask-R-CNN identifies such relevant salient pedestrians. Specifically, the driver is sober, but in the driving scene, there are relevant pedestrians, and therefore, an average attention is needed on the part of the driver who, even if sober, must pay attention to the tracked relevant pedestrians in the driving scene. No further automation and control mechanism is envisaged in the pipeline proposed here.

The acoustic signal system is managed by the STA1295A Accordo5 Audio sub-system, which hosts the ICP software implementation [38,39].

## 4. Experimental Results

Each of the implemented sub-systems have been validated. Regarding the Intelligent Soft-Sensing System—we have tested the designed pipeline as follows. We have used an ad-hoc dataset with about 9000 detections made by using the VOC GHT25S sensor emulating a driving scenario as schematized in Figure 2. The sensing device was calibrated on such internals of a car and for two different subjects, representing the possible car drivers—one male and the other female. For different subjects, the sensing system had to be calibrated by performing breath data sampling to be used in the downstream 1D-CNN deep network fine tuning. For each measurement session, the recruited driver performed 5 min of sensor acquisition, spaced approximately 1 m both in a scenario where he/she had drunk enough alcohol to exceed the allowed BAC and in scenarios where no substance containing alcohol was ingested in the past 3 h.

Some details on the sampling test dataset setup are as follows. We equipped a steering wheel with the sensor described in this paper. Simultaneously, we emulated driving scenes with public datasets (such as CamVid, DH1FK, etc.) in a monitor in front of the driver. In the following Figure 7, our steering was equipped with the GHT25S sensor together with the MCU system based on STA1295 Accordo5.

We acquired the data with the subject-driver sitting in front of the sensing-enhanced steering wheel, both in a sober condition and not.

The so-recruited dataset has been split as follows: 70% for training while remaining and 30% for testing and validation. A k-fold (k = 3) cross-validation has been applied to reduce the over-fitting issue.

We performed our simulation on an Intel multicore server with NVIDIA GPU RTX 2080 with 8 GB of video memory. We tested several further deep architectures to perform a robust benchmarking with respect to our proposed solution.

We checked a classical machine learning pipeline with a fully connected multi-layer network (FCN) (hidden layer with 50 neurons), embedding a leakyRelu actiavtion batch normalization layer as well as an LSTM vanilla-based solution with 200 hidden cells. For the deep architecture, we applied a classical SGD learning algorithm, initial learning rate of 0.01, L2 regularization and adams optimization. Moreover, we checked a Support Vector Machine (SVM) approach for classifying the sensing input data. As input data, we used the normalized data coming from the sampling session mentioned in the previous section, specifically: ppm, temperature, Rsense, humidity and Rair for each car driver breath-sensing acquiring session. The following Table 1 reports the experimental benchmark results for the tested intelligent approaches, best results are highlighted in bold.

After that, we tested the Intelligent Pedestrian Tracking system on CamVid, retrieving a promising test set performance mIoU of 0.695, which is comparable with similar deep architectures [29,47] as reported in the following Table 2.

## 5. Conclusions and Future Works

The authors proposed an intelligent combination system able to perform a very fast discrimination of the driver’s sobriety in a challenging automotive scenario, i.e., the ones embedding pedestrians. The proposed method requires the car to embed a sustainable VOC sensor (we tested GHT25S) suitable to monitor the car subject’s breath while driving. For this purpose, an ad-hoc GHT25S-based bio-sensing system has been implemented and embedded in the car steering. The validations carried out confirmed that a VOC data buffering of 50/60 s are enough to allow the correct discrimination of the driver’s sobriety, considering this temporal buffering, starting from the moment in which the effect of the ingested alcohol is evident in the organic compounds embedded in the breath of the subject driving. The parallel proposed intelligent pedestrian tracking systems show very promising results. The accuracy of the full system reached high levels due to the innovative intelligent approach applied to post-processing of the breath-sensing data coming from the driver. The whole pipeline is ongoing to be ported over the STA1295A Dual-Core ARM A7 ACCORDO 5 plus SPC58x MCUs platform, provided by STMicroelectronics and in which a custom YOCTO Linux embedded operating framework is running. This system is equipped with a 3D accelerated graphics core [38,39]. The proposed sensing system was calibrated to selectively recognize the presence of alcohol in the driver’s breath. In fact, the downstream deep architectures have been trained on the data of the driver; therefore, there is a certain selectivity also correlated to the data relating to the breath of the subject on which the system has been trained. Currently, in our tests, only the driver can enable an alcohol detection due to the relative proximity to the sensor, while the other subjects in the car were not able to activate the sensor although they had drunk alcoholic substances even at significant concentrations. Our tests were performed by emulating the passenger compartment of the car and putting the sober driver and a close subject who had been drinking alcohol. However, if the passenger compartment of the vehicle becomes saturated with alcohol (or in the absence of ventilation in the passenger compartment) due to the presence of several subjects who have drunk quantities exceeding the permitted thresholds (sensor calibration), the designed sensing system could equally raise an alcohol alert, as the whole environment would be saturated with alcohol in the air. On this item, we are working on an algorithm, which, besides recognizing the presence of alcohol selectively, is also able to robustly recognize a sort of “breath fingerprint” of the driver to uniquely calibrate the sensing system. Studies are underway using the recent transformer-based architectures [40].

This study reports preliminary results of the implemented pipeline for a car driver breath-based alcohol sensing system through a simple VOC sensor. Further investigations are needed to improve the dataset, the testing conditions, etc. Future works aim to extend the proposed pipeline to a large dataset in the aim of a clinical study in which the embedding features of alcohol in the breath can be deeply analyzed.

## Figures and Tables

**Figure 1 sensors-22-00674-f001:**
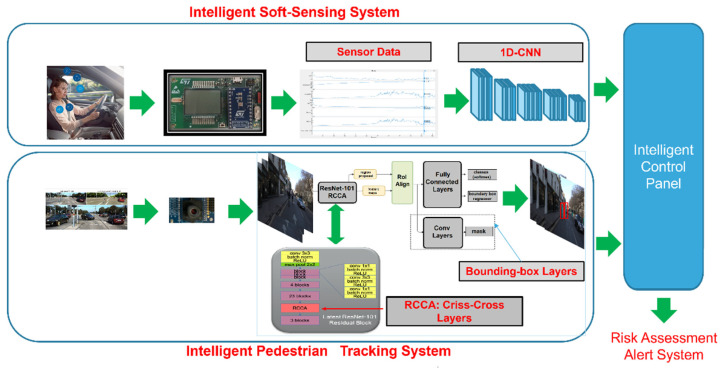
The proposed intelligent car driver assisting pipeline.

**Figure 2 sensors-22-00674-f002:**
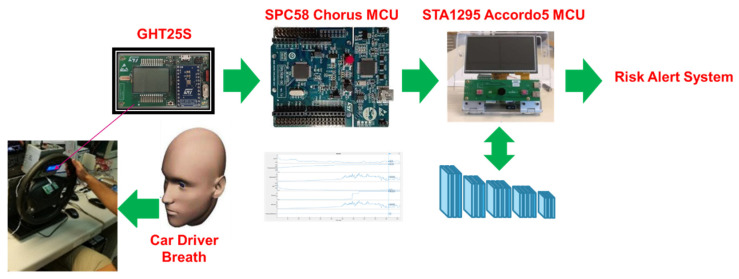
Schematic overview of the proposed GHT25S-based sensing system.

**Figure 3 sensors-22-00674-f003:**
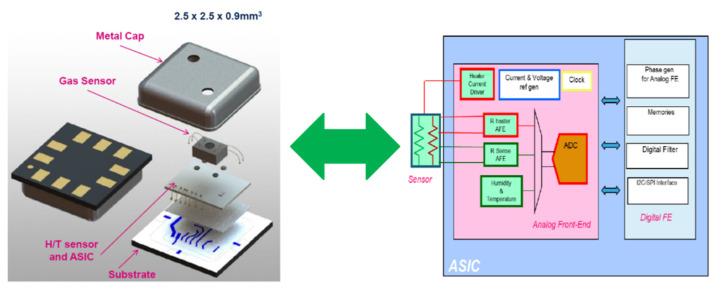
Schematic internal overview of the GHT25S sensing device.

**Figure 4 sensors-22-00674-f004:**
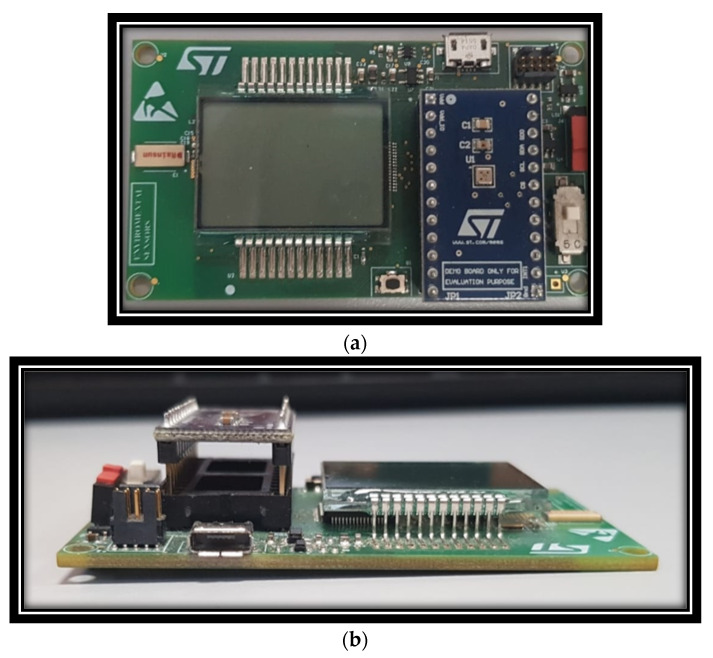
An overview of the GHT25S sensor: (**a**) frontal view; (**b**) side view.

**Figure 5 sensors-22-00674-f005:**
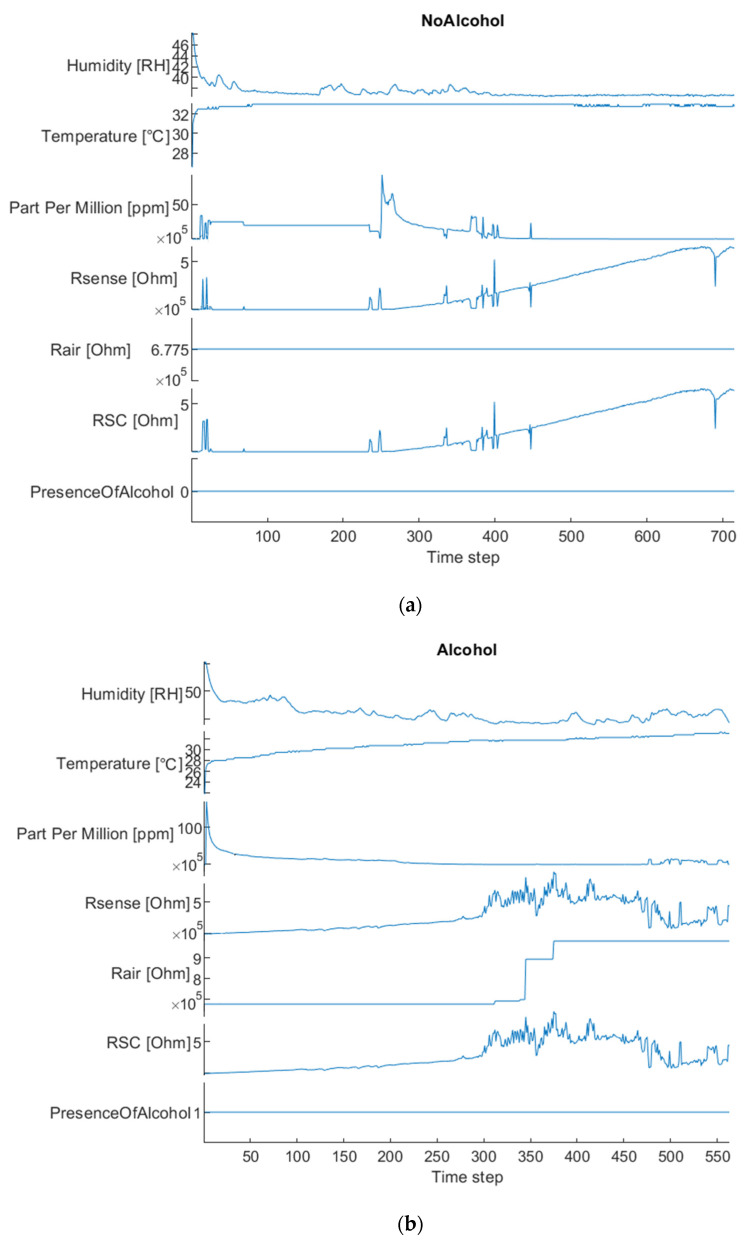
GHT25S car driver breath sampling data diagram: (**a**) sober driver; (**b**) driver who drank alcohol.

**Figure 6 sensors-22-00674-f006:**
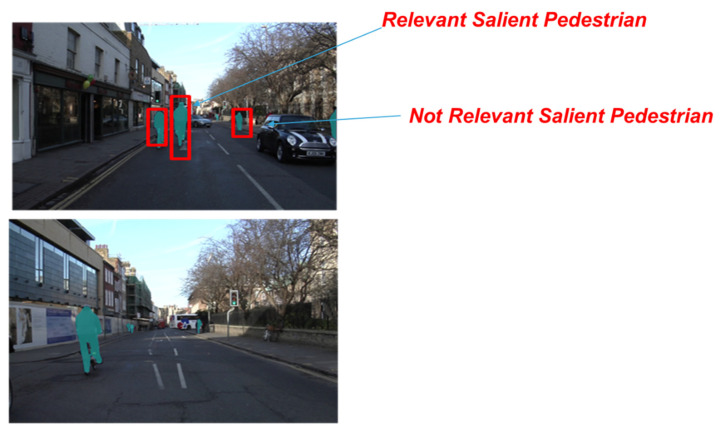
Some instances of the driving scene frames overlaid with segmented salient pedestrians in different configurations (on foot, by bicycle, etc.). In red are the predicted bounding-boxes and related relevant/not relevant pedestrian assessment.

**Figure 7 sensors-22-00674-f007:**
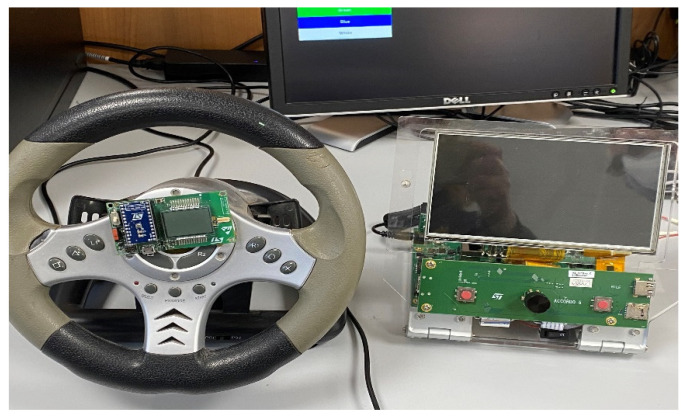
The testing setup of the proposed sensing system.

**Table 1 sensors-22-00674-t001:** Car driver alcohol detection by soft-sensing system: experimental results.

Model	Accuracy	Sensitivity	Specificity
1D-CNN	**0.9823**	**0.9811**	**0.9839**
FCN	0.9775	0.9802	0.9755
SVM	0.9500	0.9800	0.9700
LSTM	0.8952	0.8905	0.9001

**Table 2 sensors-22-00674-t002:** Intelligent pedestrian tracking system: experimental results (CamVid dataset).

Method	Intelligent Pedestrian Tracking System mIoU
Proposed	**69.50%**
Faster-R-CNN (ResNet-50 backbone)	53.95%
Mask-R-CNN with ResNet-101 backbone w/o Criss-Cross RCCA	63.96%

## Data Availability

Data sharing not applicable.

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
