# Peer review of "Advanced eNose-Driven Pedestrian Tracking Pipeline for Intelligent Car Driver Assisting System: Preliminary Results"

_sensors, 2022, doi:10.3390/s22020674_

Round 1
Reviewer 1 Report
This article presents an intelligent system which is able to perform a fast discrimination of the driver's sobriety in challenging scenarios detecting pedestrians, in a car. A VOC sensor monitors the driver's breath while driving.
The english has to be slightly corrected by a native speaker.
I have several questions:
1) Page 10, line 306, It is not quite clear the conditions of High Risk and Medium-Low Risk levels.
2) What happens if the passenger has been drunk and not the driver ?
3) Does the car take control if there are pedestrians spotted by the pedestrian tracking system and the driver is drunk (or the sensor is indicating he/she is drunk) ? or there is only an acoustic signal.
4) Can you explain in detail the experience with a man and a woman and sensor acquisition and calibration.
Thanks for this very interesting research.
Author Response
We would like to thank the reviewer for his valuable comments. All the reviewers' concerns have been addressed and suggestions have been applied in the revised version of the manuscript. Please see the attachment for the details about the response to the reviewer's comments.

Reviewer 2 Report
My first consideration is the following: the article presents "preliminary results" of the study, and I am afraid that preliminary results do not correspond to the format of a full article.
Another issue: according to the title of the paper the authors presents "Pedestrian Tracking Pipeline" driven by electronic nose. But in the abstract the core of the paper is formulated as "a full deep pipeline for the identification, monitoring and tracking of the salient pedestrians combined with an intelligent electronic alcohol sensing system to proper assess the physiological status of the driver". So the question is what the authors present: a pedestrian tracking pipeline or a system to proper assess the physiological status of the driver.
Unfortunately the main contributions of the paper are not formulated, and the paper itself gives no opportunity to conclude what they are.
So the very first issues create the background for overall opinion - the paper needs major revision.
The main concerns on the content and the presentation of the article are the following:
The Introduction is awaited to give the motivation for the study: why this problem is important, what approaches exist and the main idea to improve the existing solutions.
Related work section unfortunately looks like description of sensors used for alcohol detection and very short review of deep learning approaches for detection and monitoring of pedestrians. The analysis of advantages and drawback of the existing techniques is needed in this section.
The Methods and Material section gives extra attention to the hardware description, but the developed methods are not adequately presented. The section The Intelligent Pedestrians Tracking System is adapted text from reference [34], and unfortunately helps a little in understanding the technique used.
Also the test of the paper contains a lot of unexplained abbreviations and typos. For example:
line 102: "sof-sensing device"
line 343: "with simila deep"
line 364: "Dual-Crore ARM"
line 306: Why "Detection of “Sober Driver” AND relevant salient pedestrians" is of "High Risk Level"?
Figure 1, 3, 5 are too small to be easily read.
Author Response

(The authors gave the same response as above.)

Reviewer 3 Report
In this article, the authors consider of autonomous driving safety, proposed a full deep pipeline with The Intelligent Soft-Sensing System and The Intelligent Pedestrians Tracking System for the identification, and an Intelligent Control panel was proposed to combine the data from above with neural networks, and well conclusions were obtained.
The following are some of the improvements which the author may need to prove in this paper.
- The images used in this paper are not intuitive enough to reflect the actual process, and the construction of the neural network lacks an intuitive flow chart. For example, in Figure 1, the ResNet-101 is unclear to see, the rest part of this figure are also missing labels. Could authors clearly mark each part, and make sure that each part of the image is clearly defined?
- The existing neural network structure is utilized in this paper, and only the parameters are modified to fit the data, which means that this paper is not innovative enough and does not present enough constructive ideas. Could the authors try to incorporate more innovations in the structure of neural networks?
- The Intelligent Control Panel is the core part of this paper, but it is extremely vague in this paper, missing many technical details and processes, and lacking a well-formed judgment standard to deal with the data input from The Intelligent Soft-Sensing System and The Intelligent Pedestrians Tracking System, resulting in unconvincing conclusions. Can the authors describe in detail the role played by The Intelligent Control Panel in the overall experiment, and give a reasonable explanation for doing so?
- There seems to be some problems on the grammar used in this paper, resulting in some parts of the description not being clear enough. The layout of the last row of Figure 2 is confusing. All in all, there are a lot of details in the article that need to be revised. Can the author refine the grammatical and typographical details of the article to make it more readable?
- Fewer methods were used for comparison, the performance of only one data set is compared under one parameter setting, and the presentation of the experimental results is not intuitive enough. As we can see in the table 1, the final conclusion has no significant advantage over the previous method. Can the authors add more experiments to demonstrate the superiority of this method?
Author Response

(The authors gave the same response as above.)

Round 2
Reviewer 3 Report
The authors have addressed all my comments, and the manuscript can be accepted.